# Lobar emphysema ratio of more than 1% in the lobe with lung cancer as poor predictor for recurrence and overall survival in patients with stage I non-small cell lung cancer

**Jeong Pyo Lee**[1], **Jae Bum Na**[1], **Ho Cheol Choi**[1], **Hye Young Choi**[1], **Ji Eun Kim**[1], **Hwa Seon Shin**[1], **Jung Ho Won**[1], **Sa Hong Jo**[1], **Seok Jin Hong**[1], **Won Jeong Yang**[1], **Yang Won Kim**[1], **Byeong Ju Koo**[1], **In Seok Jang**[2], **Mi Jung Park**[1] *

**1** Department of Radiology, Gyeongsang National University School of Medicine, Gyeongsang National University Hospital, Jinju, Korea, **2** Department of Cardiothoracic Surgery, Gyeongsang National University School of Medicine, Gyeongsang National University Hospital, Jinju, Korea

* pichola@naver.com

## Abstract

### Background

The purpose of this study was to examine the relationship between the lobar emphysema ratio (LER) and tumor recurrence and survival in patients with stage I non-small cell lung cancer (NSCLC).

### Methods

We enrolled 258 patients with surgically proven stage I NSCLC. These patients underwent noncontrast chest CT, and pulmonary lobe segmentation and lobar emphysema quantification were performed using commercially available software. We assessed the LER in the lobe with lung cancer. We divided the patients into two groups according to the LER, and the cut-off value was 1. Furthermore, we analyzed the disease-free survival of high LER and other clinical factors after surgical resection.

### Results

The 258 patients were divided into two groups: low LER (n = 195) and high LER (n = 63). The right upper lobe was the most frequent location in lung cancer and the most severe location in emphysema. In the Kaplan–Meier curve, high LER showed a significantly lower disease-free survival (8.21 ± 0.27 years vs 6.53 ± 0.60 years, p = 0.005) and overall survival (9.56 ± 0.15 years vs. 8.51 ± 0.49 years, p = 0.011) than low LER. Stage Ib (2.812 [1.661–4.762], p<0.001) and high LER (2.062 [1.191–3.571], p = 0.010) were poor predictors for disease-free survival in multivariate Cox regression analysis. Stage Ib (4.729 [1.674–13.356], p = 0.003) and high LER (3.346 [1.208–9.269], p = 0.020) were significant predictors for overall survival in multivariate Cox regression analysis.

**Data Availability Statement:** All relevant data are within the paper and its Supporting Information files.

**Funding:** The author(s) received no specific funding for this work.

**Competing interests:** The authors have declared that no competing interests exist.

## Conclusion

A LER of more than 1% in the lobe with lung cancer is a poor predictor for cancer recurrence and overall survival in patients with stage I NSCLC.

## Introduction

Chronic obstructive pulmonary disease (COPD) is the main risk factor for lung cancer. Approximately 1% of patients with COPD progress to lung cancer every year [1]. COPD can be diagnosed with pulmonary function tests and pathologic studies. Previous studies suggest that emphysema assessed by CT is well correlated with airflow limitation in pulmonary function tests and emphysematous areas in pathologic specimens [2,3]. Furthermore, chest CT has been widely used in patients with lung cancer for screening and risk factor stratification and recurrence after pulmonary resection [4–8]. The size of lung cancer, nodal and distant metastasis, and smoking history are poor predictors for lung cancer recurrence [9].

Emphysema is another important prognostic factor for lung cancer recurrence [4–6,10–16]. However, we should carefully consider several factors, such as location and threshold value, to quantify emphysema using CT densitometry. Initially, the whole lung was quantitatively measured by CT, but the lobe-based quantification of emphysema is possible with technological advances. The pulmonary lobe is a fundamental component that is linked to lung cancer progression and the degree of surgical resection. As a result, we considered that lobe-based segmentation provides more physiological data to understand lung cancer than zone-based segmentation with nonanatomic division. The semiautomatic software can be helpful to extract lung volume, detect fissures and quantify lobe-based emphysema. Determining the specific threshold value of CT attenuation is important to measure the exact emphysematous volume. The -950 HU instead of -910 HU has been widely used as the cutoff value for emphysematous area. Furthermore, we must decide the specific threshold of the emphysematous ratio to divide the study population into patients with and without emphysema. The COPDGene study recommended more than 5% emphysema as patients with emphysema [17]. However, the prognostic impact of minimal emphysema has rarely been investigated in patients with lung cancer. Therefore, we lowered the threshold value of the emphysema ratio from 5% to 1% to include patients with minimal emphysema. This study investigates the prognostic role of lobe-based emphysema scores in patients with stage I non-small cell lung cancer (NSCLC).

## Methods

### Patient characteristics

In total, 480 patients with suspected lung cancer were consecutively enrolled in our hospital from January 2011 to December 2015. We collected the patients' data, such as age, sex, and smoking history. All patients underwent pulmonary function tests and chest CT preoperatively. The exclusion criteria were as follows: stage II, III, and IV NSCLC (n = 131), preexisting malignancy (n = 13), small cell lung cancer or metastatic lesion by histopathologic specimen (n = 9), history of prior pulmonary resection (n = 4), and insufficient patient records, such as unknown smoking history (n = 7). We also excluded cases with chest CT with poor image quality (n = 15) and chest CT taken in outside clinics (n = 24) because good image quality and thin section axial images are essential to precisely quantify emphysema. However, the automatic lobe segmentation of the lung parenchyma failed in some patients (n = 19) due to an

unknown software error. The stage and histologic type of lung cancer were determined using the 8th edition of the Tumor Node Metastasis classification and World Health Organization classification [18,19]. The histopathologic specimen of the 258 individuals revealed stage I NSCLC. The patients underwent chest CT in an outpatient clinic at 3- to 6-month intervals for the first 2 years and then annually thereafter. The term "disease-free survival (DFS)" was defined as the period from the time of pulmonary resection to the time of recurrence, metastasis, or death. The term "overall survival (OS)" was defined as the period from the time of lung cancer diagnosis to the time of death. Our institutional review board gave its approval for our retrospective investigation (GNU 2021-11-004). Informed consent was waived, because this study was retrospective design and patients' data were analyzed anonymously.

## Pulmonary function test

Bronchodilator spirometry was conducted utilizing a Jaeger instrument (Wurzburg, Germany) within 1 month before surgery for lung cancer. The forced vital capacity (FVC) and forced expiratory volume in 1 second (FEV1) were recorded following the recommendations of the American Thoracic Society and European Respiratory Society [20]. A FEV1/FVC ratio of less than 0.7 indicates COPD according to the guidelines of the Global Initiative for Chronic Obstructive Lung Disease [20].

## Chest CT scan

Chest CT scans were performed within 1 month of pulmonary resection. The 64-row detector CT scanner (Brilliance-64; Philips Medical Systems, Eindhoven, Netherlands) was used for emphysema quantification, and the following parameters were followed: detector configuration of 64 x 0.625 mm, tube voltage of 120 kVp, tube current of 200 mAs, pitch of 0.923, and gantry rotation time of 500 milliseconds. With full inspiration, the chest was scanned craniocaudally from the lung apex to the diaphragm level. Contrast material was not administered in this study. All axial images were reconstructed using a smooth reconstruction filter (Philips "B" filter) with a slice thickness of 1 mm and slice interval of 1 mm.

## Lobe segmentation and emphysema quantification

Pulmonary lobe segmentation and emphysema quantification were performed using the chest imaging platform extension (https://chestimagingplatform.org) of 3D SLICER (http://www.slicer.org, version 4.11). One chest radiologist with 12 years of expertise reviewed all images while blinded to the patients' clinical data. First, axial CT images were loaded in 3D SLICER software (Fig 1A). Next, the 'interactive lobe segmentation' module within the chest imaging platform extension was applied for lobe segmentation of the lung parenchyma. Three fiducial points on each fissure were manually marked on five different axial images (2nd, 4th, 6th, 8th, and 10th thoracic vertebral levels) by a radiologist. After lobe segmentation, the interactive lung label map was created, and five lobes with different color masks were displayed in axial, coronal, and sagittal images. If the lobe segmentation was accurate, the 'parenchyma analysis' module within the chest imaging platform extension was applied for emphysema quantification. After input of the interactive lung label map in the CT images, attenuation of voxels in the whole lung and each pulmonary lobe was quantified automatically. Emphysema volume was defined as the sum of voxels with attenuation values less than −950 Hounsfield units (Fig 1B). The emphysema ratio was defined as the ratio between emphysema volume and lung volume. We obtained the emphysema ratio in the whole lung and that in each pulmonary lobe. The lobar emphysema ratio (LER) was defined as the emphysema ratio in the lobe with lung

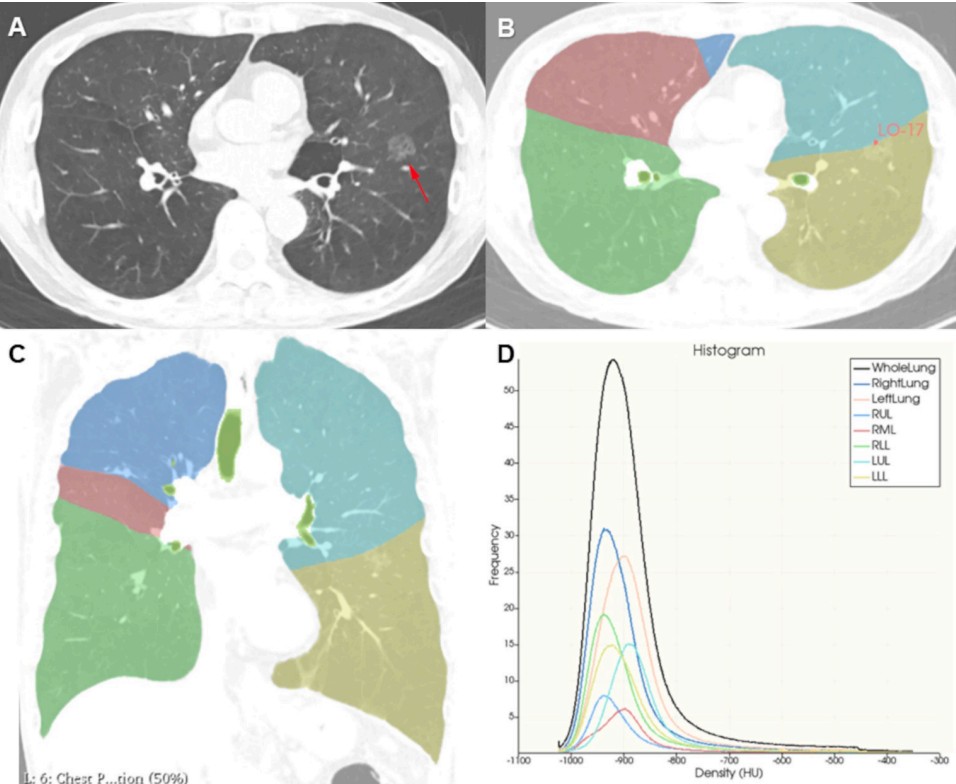

**Fig 1. The process of pulmonary lobe segmentation and emphysema quantification using 3D Slicer software.** (A) 58-year-old man with a 19 mm mixed ground glass opacity nodule (arrow) in the left lower lobe. (B, C) The pulmonary lobe is successfully extracted after marking each pulmonary fissure using the "interactive lobe segmentation" module. The right upper lobe, right middle lobe, right lower lobe, left upper lobe and left lower lobe in axial and coronal images were displayed in the blue, red, green, turquoise and yellow colors, respectively. (D) The ratios of emphysema in both lungs and each pulmonary lobe were automatically analyzed using the "parenchymal analysis" module. The lung densities in whole lung and each pulmonary lobe were presented using the density histogram analysis. The lobar emphysema ratio in the left lower lobe was 16.591, and he was confirmed to have adenocarcinoma after lobectomy.

cancer. The LER was used to divide the population into two groups, the high LER group and the low LER group, and '1' was used as the cutoff point.

## Statistical analysis

SPSS statistics software was used to conduct all statistical analyses (SPSS 21.0; SPSS Inc., Chicago, IL). Categorical variables were represented as counts with percentages, whereas continuous variables were provided as the means with standard deviations. The clinical data and LER were compared using unpaired Student's t tests for continuous variables and chi-squared tests for categorical variables. The one-way ANOVA was performed to assess the relationship between location of lung cancer and LER in high LER group. Kaplan–Meier analysis was applied to analyze the DFS and OS based on the LER, and the log-rank test was applied to compare the survival rates. The Cox proportional hazards model was applied to calculate the LER and clinicopathologic variables for DFS and OS. In a multivariate analysis, hazard ratios and corresponding 95% confidence intervals were investigated to reveal the relevant predictors in patients with stage I NSCLC. Statistical significance was set to a two-sided p value less than 0.05.

**Table 1. Patient characteristics according to the lobar emphysema ratio.**

| Characteristics | Low (<1%) lobar emphysema ratio (n = 195) | High (>1%) lobar emphysema ratio (n = 63) | p value |
|---|---|---|---|
| Age | 66.0 ± 9.6 | 70.6 ± 7.1 | 0.006 |
| Male | 108 (55.4%) | 59 (93.7%) | <0.001 |
| Smoking status | | | <0.001 |
| Non-smoker | 102 (52.3%) | 12 (19.0%) | |
| Ex- or current smoker | 93 (47.7%) | 51 (81.0%) | |
| Packyears | 16.4 ± 22.4 | 34.3 ± 29.2 | <0.001 |
| Histologic type | | | <0.001 |
| squamous cell carcinoma | 40 (20.5%) | 34 (54.0%) | |
| adenocarcinoma | 142 (72.8%) | 34 (38.1%) | |
| Others | 13 (6.7%) | 6 (7.9%) | |
| Pathologic T stage | | | 0.338 |
| T1mi | 18 (9.2%) | 1 (1.5%) | |
| T1a | 33 (16.9%) | 11 (17.5%) | |
| T1b | 60 (30.8%) | 16 (25.4%) | |
| T1c | 33 (16.9%) | 16 (25.4%) | |
| T2a | 51 (26.2%) | 19 (30.2%) | |
| FEV1/FVC ratio | 74.0 ± 8.7 | 64.4 ± 11.9 | <0.001 |
| FEV1, % pred | 85.8 ± 17.2 | 77.9 ± 18.9 | 0.002 |
| Lung resection | | | 0.528 |
| Lobectomy | 153 (78.5%) | 46 (73.0%) | |
| Sublobar resection | 39 (20.0%) | 16 (27.0%) | |
| Others | 3 (1.5%) | 0 (0%) | |
| Total emphysema ratio | 0.21 ± 0.42 | 4.35 ± 3.28 | <0.001 |
| Lobar emphysema ratio in lobe with cancer | 0.14 ± 0.21 | 5.16 ± 4.74 | <0.001 |

## Results

### Clinical characteristics

The 258 patients with stage I NSCLC were divided into the high LER group and the low LER group. Table 1 summarizes the differences in patient characteristics between the two groups. The proportion of males (93.7% vs. 55.4%, p<0.001), current or ex-smokers (81.0% vs. 44.7%, p<0.001), and duration of smoking (34.3 ± 29.2 vs. 16.4 ± 22.4 pack-years, p<0.001) were significantly higher in the high LER group. Regarding the histopathologic type, the proportion of squamous cell carcinoma (54.0% vs. 20.5%, p<0.001) was significantly higher in the high LER group. Alternatively, the ratio of adenocarcinoma (72.8% vs. 38.1%, p<0.001) was significantly higher in the low LER group. Regarding TNM stage, we found no significant difference between the two groups. In the pulmonary function test, the $FEV_1/FVC$ ratio (64.4 ± 11.9 vs. 74.0 ± 8.7) was significantly lower in the high LER group.

### Location of lung cancer and lobar emphysema ratio

The most common location of stage I NSCLC was the right upper lobe (28.3%), followed by the left upper lobe (26.4%), right lower lobe (22.9%), left lower lobe (15.1%), and right middle lobe (7.4%). In the high LER group, we examined which lobe had the most severe LER. The LER was the highest in the right upper lobe (6.24 ± 4.44), followed by the left upper lobe (5.69 ± 6.18), left lower lobe (5.24 ± 5.16), right lower lobe (3.73 ± 2.71), and right middle lobe (1.90 ± 0.75) (Table 2). We analyzed the relationship between LER and the location of lung

Table 2. Relationship between location of lung cancer and lobar emphysema ratio in the high LER group.

| Pulmonary lobe | Emphysema score in lobe with lung cancer | Cases (n = 63) |
|---|---|---|
| Right upper lobe | 6.24 ± 4.44 | 18 |
| Right middle lobe | 1.90 ± 0.75 | 3 |
| Right lower lobe | 3.73 ± 2.71 | 14 |
| Left upper lobe | 5.69 ± 6.18 | 17 |
| Left lower lobe | 5.24 ± 5.16 | 11 |

cancer. In high LER group, the mean emphysema ratio in the lobe with lung cancer (5.16 ± 4.74) was not significantly different from that in other lobes without lung cancer (4.08 ± 3.15, p = 0.135).

### Factors that affect disease-free survival and overall survival

To investigate the prognostic role of emphysema after complete cancer removal in patients with stage I NSCLC, we examined the DFS of high LER and other clinical factors. The high LER group showed significantly lower DFS than the low LER group (8.21 ± 0.27 years vs 6.53 ± 0.60 years, p = 0.005; Fig 2A) in the Kaplan–Meier curve. In addition, the high LER group had significantly worse OS than the low LER group. (9.56 ± 0.15 years vs. 8.51 ± 0.49 years, p = 0.011; Fig 2B).

Stage Ib (2.887 [1.704–4.888], p<0.001), high total emphysema ratio (1.766 (1.014–3.076), p = 0.045), and a high LER (2.150 [1.240–3.728], p = 0.006) were related to DFS in univariate Cox regression analysis. Stage Ib (2.812 [1.661–4.762], p<0.001) and high LER (2.062 [1.191–3.571], p = 0.010) were significant predictors for DFS in multivariate Cox regression analysis (Table 3).

Stage Ib (5.180 [1.840–14.582], p = 0.002) and high LER (3.843 [1.377–10.720, p = 0.010) were related to OS in univariate Cox regression analysis. Stage Ib (4.729 [1.674–13.356], p = 0.003) and high LER (3.346 [1.208–9.269], p = 0.020) were significant predictors for OS in multivariate Cox regression analysis (Table 4).

### Discussion

We found that a LER of more than 1% in the lobe with lung cancer at baseline was a poor predictor of DFS and OS in patients with stage I NSCLC. The upper lobe was the most frequent

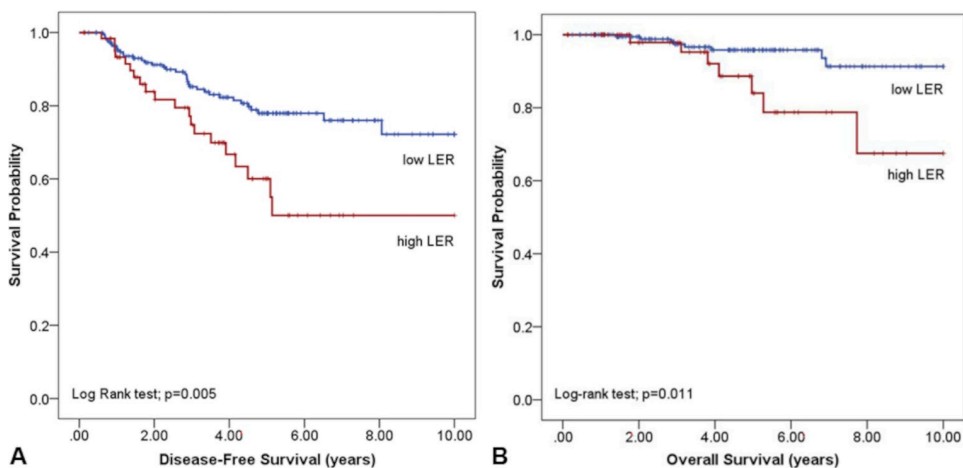

Fig 2. Disease-free survival (A) and overall survival (B) in high LER and low LER group.

**Table 3. Univariate and multivariate analysis of significant predictors for the disease-free survival in patients with stage I NSCLC.**

| Variable | Univariate | | Multivariate | |
|---|---|---|---|---|
| | HR (95% CI) | p value | HR (95% CI) | p value |
| Smoking | 1.189 (0.700–2.020) | 0.522 | | |
| Pathologic stage (Ib) | 2.887 (1.704–4.888) | <0.001 | 2.812 (1.661–4.762) | <0.001 |
| FEV1/FVC <0.7 | 1.181 (0.694–2.012) | 0.540 | | |
| Total emphysema ratio (>1) | 1.766 (1.014–3.076) | 0.045 | | |
| Lobar emphysema ratio (>1) | 2.150 (1.240–3.728) | 0.006 | 2.062 (1.191–3.571) | 0.010 |

One-way ANOVA; p = 0.438

location in lung cancer. The upper lobe had a greater emphysema score than the middle and lower lobes in high LER group.

In our study, DFS was significantly lower in patients with emphysema than in those without emphysema. Similar studies have investigated the predictive role of emphysema in NSCLC patients. Bishawi et al. demonstrated that a high regional emphysema score using a visual scale showed a poor long-term survival rate in early lung cancer [21]. Visual assessment is a subjective method that leads to interobserver error. Because lung cancer may be found when emphysema is analyzed using a visual scale. Amaza et al. analyzed the concordance rate according to the presence of emphysema between visual and quantitative analysis, and the concordance rate ranged from 56 to 61 depending on the study population [22]. Therefore, CT quantification is crucial for analyzing emphysema severity to reduce interobserver variation. However, some studies using CT quantitative analysis reported inconsistent results for the association between lung cancer and emphysema. Some studies [23,24] reported that emphysema was a nonsignificant risk factor for early lung cancer, but other studies [16] reported that those patients with high emphysema scores showed high mortality. We speculated that the different CT protocols and study populations affected these controversial results.

Density histogram analysis by CT is used to establish a reference value of density between normal lung parenchyma and emphysema. Previous studies revealed that NSCLC patients with high emphysema scores showed a poor survival rate [12,13]. These studies used the old version of CT with 10 mm slice thickness, and an attenuation threshold lower than -910 HU was defined as an emphysematous area on chest CT [12,13]. Other studies revealed that an area less than -950 HU was correlated with the emphysematous area on pathologic specimens [3,25]. For emphysema quantification, we applied -950 HU as a cutoff value with 1 mm slice thickness on 64-slice multidetector CT.

Selecting the optimal LER threshold to differentiate between patients with and without emphysema is also important. The COPDGene study recommended that 5% is the cutoff value of LER to divide the patients into two groups [17]. However, lung cancer patients with severe emphysema can be excluded from surgery in a real-world clinical setting because these patients

**Table 4. Univariate and multivariate analysis of significant predictors for the overall survival in patients with stage I NSCLC.**

| Variable | Univariate | | Multivariate | |
|---|---|---|---|---|
| | HR (95% CI) | p value | HR (95% CI) | p value |
| Smoking | 1.568 (0.557–4.416) | 0.394 | | |
| Pathologic stage (Ib) | 5.180 (1.840–14.582) | 0.002 | 4.729 (1.674–13.356) | 0.003 |
| FEV1/FVC <0.7 | 1.550 (0.560–4.287) | 0.398 | | |
| Total emphysema ratio (>1) | 1.934 (0.657–5.694) | 0.231 | | |
| Lobar emphysema ratio (>1) | 3.843 (1.377–10.720) | 0.010 | 3.346 (1.208–9.269) | 0.020 |

can have increased long-term mortality and postoperative complication [4]. Even though the threshold of the emphysema ratio was lowered from 5% to 1% in our study, the patients with more than 1% of emphysema showed poor DFS and OS in patients with stage I NSCLC. This finding emphasizes the significance of minimal emphysema in the lobe with early lung cancer.

Another technical issue for emphysema quantification is selecting the extent of emphysema quantification, including whole lung segmentation, zone-based segmentation, and lobe-based segmentation. Previous studies showed that emphysema was related to mortality in patients with lung cancer, but they quantified emphysema in the whole lung. Emphysema is not evenly distributed, so regional emphysema quantification may offer valuable information to understand the association with lung cancer. Previous research demonstrated a relationship between the zone with lung cancer and the regional emphysema score in that zone. Previous studies have shown that the regional emphysema score in the zone with lung cancer is related to the location [15,21,26] and recurrence [16]. The zone-based segmentation is not anatomical but the virtual plane, which horizontally divides the whole lung into three zones. Bae et al. reported that LER was high in the lobe with lung cancer [27]. We believe that lobe-based segmentation is appropriate for understanding the relationship between lung cancer and emphysema.

Previous studies using visual semiquantitative and lobe-based quantitative assessment showed that the regional emphysema score was significantly greater in the lobe with lung cancer than in other lobes without lung cancer [15,27]. However, we found no significant difference in LER between the two groups. We speculated that the inconsistent result is due to the difference in the study population. We enrolled patients with surgically confirmed NSCLC, so inoperable cases, such as poor pulmonary function or severe emphysema, were excluded from our study.

Although airflow obstruction has been known as an independent risk factor for lung cancer [23,24], its prognostic significance is still controversial in patients with early-stage lung cancer. Lopez et al. reported that airflow obstruction is a poor prognostic factor for 2-year survival in stage I lung cancer [28]. However, Ueda et al. reported that airflow obstruction is not a significant poor prognostic factor for 5-year OS and DFS in patients with lung cancer, mostly early-stage lung cancer [13], these findings are consistent with our results. Further research are required to verify the prognostic value of airflow obstruction in patients with lung cancer.

Smoking status is a well-known poor prognostic factor for survival in patients with lung cancer [29,30], furthermore smoking cessation after diagnosis lowers the mortality in patients with early-stage lung cancer [31]. In contrast, our results indicated that smoking status is not a significant prognostic factor for OS and DFS. Compared to previous large-scale investigations, the number of study population and mortality rate were low in our study. These factors might affect our study's results.

Our study has the following limitations. First, our study is a retrospective methodology, and most patients have no or mild emphysema, which leads to selection bias. Second, lobe segmentation for emphysema quantification was performed by one radiologist. The fiducial marker was manually placed in the fissure, so interobserver error is inevitable when quantifying LER. Third, there is no CT standard protocol for emphysema quantification. International consensus is needed to acquire reproducible data. Fourth, although we did not gather information on diffusion capacity in all patients, it may be a complicating factor for survival in patients with lung cancer. Patients with normal PFT in our institution are optional to undergo the diffusion capacity test.

## Conclusions

In conclusion, emphysema of more than 1% in the lobe with lung cancer was a poor predictor for disease-free survival and overall survival in stage I NSCLC. Therefore, a close follow-up

study should be performed in cases of minimal emphysema in the lobe with lung cancer. In addition, semiautomatic CT software can be a useful tool to assess lobar segmentation and emphysema quantification.

## Supporting information

**S1 File.**
(XLSX)

## Author Contributions

**Conceptualization:** Ho Cheol Choi, Yang Won Kim.

**Data curation:** Jeong Pyo Lee, Byeong Ju Koo, In Seok Jang, Mi Jung Park.

**Formal analysis:** Ji Eun Kim, Seok Jin Hong, In Seok Jang, Mi Jung Park.

**Investigation:** Ho Cheol Choi, Ji Eun Kim, In Seok Jang, Mi Jung Park.

**Methodology:** Jeong Pyo Lee, Sa Hong Jo, Mi Jung Park.

**Project administration:** Hye Young Choi, Byeong Ju Koo.

**Resources:** Jung Ho Won, Mi Jung Park.

**Software:** Hye Young Choi, Jung Ho Won, Sa Hong Jo, Won Jeong Yang.

**Supervision:** Jae Bum Na.

**Validation:** Hwa Seon Shin, Yang Won Kim.

**Visualization:** Hwa Seon Shin, Seok Jin Hong, Won Jeong Yang.

**Writing – original draft:** Jeong Pyo Lee, Mi Jung Park.

**Writing – review & editing:** Jeong Pyo Lee, Mi Jung Park.

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
