## [Decision Letter · Decision Letter 0]

12 Dec 2022

PONE-D-22-30040Lobar emphysema ratio of more than 1% in the lobe with lung cancer as poor predictor for recurrence and overall survival in patients with stage I non-small cell lung cancerPLOS ONE

Dear Dr. Park,

Thank you for submitting your manuscript to PLOS ONE. After careful consideration, we feel that it has merit but does not fully meet PLOS ONE’s publication criteria as it currently stands. Therefore, we invite you to submit a revised version of the manuscript that addresses the points raised during the review process.

We look forward to receiving your revised manuscript.

Kind regards,

Kartikeya Rajdev, MD

Academic Editor

PLOS ONE

Journal Requirements:

3. Please upload a new copy of Figure 1b as the detail is not clear. Please follow the link for more information:

https://blogs.plos.org/plos/2019/06/looking-good-tips-for-creating-your-plos-figures-graphics/
https://blogs.plos.org/plos/2019/06/looking-good-tips-for-creating-your-plos-figures-graphics/

Reviewers' comments:

Reviewer's Responses to Questions

**Comments to the Author**

1. Is the manuscript technically sound, and do the data support the conclusions?

Reviewer #1: Yes

Reviewer #2: Yes

Reviewer #3: Yes

2. Has the statistical analysis been performed appropriately and rigorously? 

Reviewer #1: I Don't Know

Reviewer #2: Yes

Reviewer #3: Yes

3. Have the authors made all data underlying the findings in their manuscript fully available?

Reviewer #1: Yes

Reviewer #2: Yes

Reviewer #3: Yes

4. Is the manuscript presented in an intelligible fashion and written in standard English?

Reviewer #1: Yes

Reviewer #2: Yes

Reviewer #3: Yes

5. Review Comments to the Author

Reviewer #1: Very well written. Very few studies have been published focusing on the correlation between lobar emphysema and stage I non-small cell lung cancer. There are a few studies on a similar topic, and this study is consistent with published literature.

Reviewer #2: Well done study. Conclusion was expected as patients with more severe emphysema will have poor outcome in general. Given the higher rates of active smoker and worse Fev1 in the HER group, they could potentially be a confounding factor when looking at DFS and OS. Since we are discussing emphysema in this study it would have been better if diffusion capacity (DLCO) was also reported for each category to look for another potential confounding factor.

Reviewer #3: Very nicely done study.

Manuscript is well written and easy to follow.

I had one comment

1)Did the authors look at degree of airflow obstruction(FEV1%) as a predictor of decreased DFS or OS? It will add to the findings of references 21,23,24 in the text.

6. PLOS authors have the option to publish the peer review history of their article (what does this mean?). If published, this will include your full peer review and any attached files.

Reviewer #1: No

Reviewer #2: No

Reviewer #3: No

---

## [Author Response · Author response to Decision Letter 0]

8 Jan 2023

Title Page (Page 1, Line 9, 12)

I added the information about affiliation “Gyeongsang National University Hospital” 

Title Page (Page 1, Line 14)

I changed the word ‘pichola’ to ‘MJP’.

; We uploaded our minimal data set as supporting information file. 

3. Please upload a new copy of Figure 1b as the detail is not clear. Please follow the link for more information:

; Figure 1b is a multipanel image. So I changed figure 1b into figure 1b,c,d for better understanding. And I added some information in the figure legend about figure 1b,c,d.

I added the word “in axial and coronal images” in Line 131, Page 6.

I added the following sentence in Line 134-135, Page 6

“The lung densities in whole lung and each pulmonary lobe were presented using the density histogram analysis. The lobar emphysema ratio in the left lower lobe was 16.591, and he was confirmed to have adenocarcinoma after lobectomy.”

Reviewer #1: Very well written. Very few studies have been published focusing on the correlation between lobar emphysema and stage I non-small cell lung cancer. There are a few studies on a similar topic, and this study is consistent with published literature.

; Thank you for your nice comments.

Reviewer #2: Well done study. Conclusion was expected as patients with more severe emphysema will have poor outcome in general. Given the higher rates of active smoker and worse Fev1 in the HER group, they could potentially be a confounding factor when looking at DFS and OS. Since we are discussing emphysema in this study it would have been better if diffusion capacity (DLCO) was also reported for each category to look for another potential confounding factor.

; I agree with your opinion. I reviewed some related articles, added the following paragraph in the discussion section. (Line 260-271 Page 14)

“Although airflow obstruction has been known as an independent risk factor for lung cancer [23,24], its prognostic significance is still controversial in patients with early-stage lung cancer. Lopez et al. reported that airflow obstruction is a poor prognostic factor for 2-year survival in stage I lung cancer [28]. However, Ueda et al. reported that airflow obstruction is not a significant poor prognostic factor for 5-year OS and DFS in patients with lung cancer, mostly early-stage lung cancer [13], these findings are consistent with our results. Further research are required to verify the prognostic value of airflow obstruction in patients with lung cancer. 

Smoking status is a well-known poor prognostic factor for survival in patients with lung cancer [29,30], furthermore smoking cessation after diagnosis lowers the mortality in patients with early-stage lung cancer [31]. In contrast, our results indicated that smoking status is not a significant prognostic factor for OS and DFS. Compared to previous large-scale investigations, the number of study population and mortality rate were low in our study. These factors might affect our study's results. 

”

I also mentioned the diffusion capacity in the limitation section. (line 277-279, Page 14, 15)

“Fourth, although we did not gather information on diffusion capacity in all patients, it may be a complicating factor for survival in patients with lung cancer. Patients with normal PFT in our institution are optional to undergo the diffusion capacity test.”

Reviewer #3: Very nicely done study.

Manuscript is well written and easy to follow.

I had one comment

1) Did the authors look at degree of airflow obstruction(FEV1%) as a predictor of decreased DFS or OS? It will add to the findings of references 21,23,24 in the text.

; 

Thank you for your comments. Previous studies indicated that the prognostic role of airflow obstruction (FEV1%) is controversial in patients with lung cancer. 

The prognostic impact of airflow obstruction is uncertain in reference 21. The reference of 23, 24 revealed the association between airflow obstruction and risk of lung cancer. 

I reviewed the other related articles and added this paragraph in the discussion section (Line 260-266, Page 14). 

“Although airflow obstruction has been known as an independent risk factor for lung cancer [23,24], its prognostic significance is still controversial in patients with early-stage lung cancer. Lopez et al. reported that airflow obstruction is a poor prognostic factor for 2-year survival in stage I lung cancer [28]. However, Ueda et al. reported that airflow obstruction is not a significant poor prognostic factor for 5-year OS and DFS in patients with lung cancer, mostly early-stage lung cancer [13], these findings are consistent with our results. Further research are required to verify the prognostic value of airflow obstruction in patients with lung cancer.” 

Finally I corrected some minor error.

1. I changed the word from “nonesmall” to “non-small” in the reference 21, Line 339

Bishawi M, Moore W, Bilfinger T. Severity of emphysema predicts location of lung cancer and 5-y survival of patients with stage I nonesmall cell lung cancer. J Surg Res. 2013;184: 1–5. doi:10.1016/j.jss.2013.05.081



Bishawi M, Moore W, Bilfinger T. Severity of emphysema predicts location of lung cancer and 5-y survival of patients with stage I non-small cell lung cancer. J Surg Res. 2013;184: 1–5. doi:10.1016/j.jss.2013.05.081

2. I changed the word “emphysema quantification” to “density histogram analysis” in Line 226, Page 12.

---

## [Decision Letter · Decision Letter 1]

31 Jan 2023

Lobar emphysema ratio of more than 1% in the lobe with lung cancer as poor predictor for recurrence and overall survival in patients with stage I non-small cell lung cancer

PONE-D-22-30040R1

Dear Dr. Park,

We’re pleased to inform you that your manuscript has been judged scientifically suitable for publication and will be formally accepted for publication once it meets all outstanding technical requirements.

Kind regards,

Kartikeya Rajdev, MD

Academic Editor

PLOS ONE

Reviewers' comments:

Reviewer's Responses to Questions

**Comments to the Author**

1. If the authors have adequately addressed your comments raised in a previous round of review and you feel that this manuscript is now acceptable for publication, you may indicate that here to bypass the “Comments to the Author” section, enter your conflict of interest statement in the “Confidential to Editor” section, and submit your "Accept" recommendation.

Reviewer #1: All comments have been addressed

Reviewer #2: All comments have been addressed

Reviewer #3: All comments have been addressed

2. Is the manuscript technically sound, and do the data support the conclusions?

Reviewer #1: Yes

Reviewer #2: Yes

Reviewer #3: Yes

3. Has the statistical analysis been performed appropriately and rigorously? 

Reviewer #1: I Don't Know

Reviewer #2: I Don't Know

Reviewer #3: Yes

4. Have the authors made all data underlying the findings in their manuscript fully available?

Reviewer #1: Yes

Reviewer #2: Yes

Reviewer #3: Yes

5. Is the manuscript presented in an intelligible fashion and written in standard English?

Reviewer #1: Yes

Reviewer #2: Yes

Reviewer #3: Yes

6. Review Comments to the Author

Reviewer #1: (No Response)

Reviewer #2: Thank you for addressing my questions and clarifying/updating the paper. I understand that Diffusion capacity may not be readily available on all patients.

Reviewer #3: Authors have done a great job in answering the reviewer comments. Well written.Manuscript can be accepted in its current form.No further corrections/questions.

7. PLOS authors have the option to publish the peer review history of their article (what does this mean?). If published, this will include your full peer review and any attached files.

Reviewer #1: No

Reviewer #2: No

Reviewer #3: No

---

## [Editor Report · Acceptance letter]

6 Feb 2023

PONE-D-22-30040R1 

Lobar emphysema ratio of more than 1% in the lobe with lung cancer as poor predictor for recurrence and overall survival in patients with stage I non-small cell lung cancer 

Dear Dr. Park:

I'm pleased to inform you that your manuscript has been deemed suitable for publication in PLOS ONE. Congratulations! Your manuscript is now with our production department. 

Kind regards, 

on behalf of

Dr. Kartikeya Rajdev 

Academic Editor

PLOS ONE